# Effects of Foliar Application of Uniconazole on the Storage Quality of Tuberous Roots in Sweetpotato

Ximing Xu [1,2,†], Xueping Pan [1,2,†], Heyao Zhang [1,2], Zunfu Lv [1], Jiaping Xia [3], Peng Cheng [3], Melvin Sidikie George [4], Yu Chen [5], Linjiang Pang [1] and Guoquan Lu [1,*]

1   The Key Laboratory for Quality Improvement of Agricultural Products of Zhejiang Province, Institute of Root and Tuber Crops, College of Advanced Agricultural Sciences, College of Food and Health, Zhejiang A&F University, Hangzhou 311300, China
2   Key Laboratory of Marine Fishery Resources Exploitment & Utilization of Zhejiang Province, Hangzhou 310014, China
3   Crop Research Institute, Anhui Academy of Agricultural Sciences, Hefei 230031, China
4   Crop Science Department, Njala University, Njala Campus, Private Mail Bag, Freetown 999127, Sierra Leone
5   Science and Technology Innovation Service Center of Lin'an, Hangzhou 311399, China
*   Correspondence: lugq@zafu.edu.cn; Tel.: +86-138-5719-1928
†   These authors contributed equally to this work.

**Abstract:** Uniconazole (UCZ), as a plant growth regulator, has been extensively applied in sweetpotato (*Ipomoea batatas* (L.) Lam) to increase tuberous root yield and quality. It is usually used in the production of sweetpotato by foliar spray. The post-harvest storage stage is crucial for forming the quality of the sweetpotato's tuberous root. Few studies have focused on the foliar spraying UCZ-affected storage quality of sweetpotato during pro-harvest storage. To examine the effects of foliar application of UCZ on the storage quality of tuberous root, this study mainly analyzed the influence of storage quality, with (K2 and K4) and without (K1 and K3) 100 mg·L$^{-1}$ foliar spraying of UCZ, at a storage period of normal fertilizing treatments (K1 and K2) and rich fertilizing treatments (K3 and K4), on the storage quality of three representative sweetpotato varieties (Z13, Z33 and J26). Compared to the no-use UCZ treatments, the decay rate of K2 was the lowest for any storage time. The decay rate of all the varieties was 0.0% before 45 DAS. Only the decay rate of Z33 increased to 4.4% at 60 DAS ($p < 0.05$). The dry matter rate of K2 and K4 was still higher than that of K1 during 15–60 DAS in Z13 and J26 ($p < 0.05$). UCZ foliar spraying was higher than without treatment at 30–60 DAS. In Z33, the springiness of UCZ spraying was higher than no spraying treatments at 45–60 DAS. These results indicate that foliar spraying of UCZ had no effect on the storage quality of tuberous root decreasing sharply, and it sometimes kept the quality stable.

**Keywords:** sweetpotato; uniconazole spraying; texture properties; physicochemical properties; storage





## 1. Introduction

Sweetpotato (*Ipomoea batatas* (L.) Lam) is a tuberous crop and belongs to the Convolvulaceae family. It is the seventh most staple food crop, with a $1 \times 10^{11}$ kg global production per year [1]. It has several excellent properties, including a high productivity rate in various environmental conditions [2]. Sweetpotato is a momentous crop because of its usefulness as a nutritious staple food, in medicine production, animal feed and industrial applications [3]. Compared to other crops, sweetpotatoes can be grown within fertile soil because of their highly capability of undergoing nitrogen fixation [4]. However, the improvement of sweetpotato production conditions and basic soil fertility is often enhanced, resulting in vine overgrowth during the actual production process in China [5]. In excessively grown plants, the development of stems and leaves is not in synchrony with the elongation of the tuberous roots in sweetpotato. Moreover, the significant yield reduction is attributed to the

increased deposition of photo-assimilates into the above-ground plant parts at the expense of tuberous root production.

Expanding quantities of farm chemicals are being allowed for use in Green Agriculture farming; however, owing to the high cost and unknown postharvest effect, their utilization is restricted to direct or spot sprays in a few crops. Nevertheless, uniconazole (UCZ), as a plant growth retardant of the triazole family, has been extensively applied in plants to increase tolerance and improve quality [6]. The use of UCZ has proved to be potentially beneficial in water-saving agriculture [7]. Its main function is to inhibit P450 entkaurene oxidase, which catalyzes the oxidation of oncourea to oncourea acid in the gibberellic acid (GA) biosynthesis pathway [8]. Moreover, UCZ can enhance abscisic acid (ABA) and cytokinin (CK) contents, and affect the isoprenoid biosynthetic pathway [5,9]. UCZ has been widely used in staple food crop and fruit production, such as wheat, rice, avocado, mango, litchi, pecan, and macadamia [10,11]. Foliar spraying of UCZ can reduce the competition between vegetative and reproductive growth. It was found that it can alter endogenous hormones and increase chlorophyll and the photosynthetic rate, as well as increase biomass and starch accumulation in duckweed [6]. UCZ reduced vine overgrowth and promoted photosynthates translocation from the leaves to the tuberous roots in sweetpotato production [5]. Appropriate path cross-sectional areas, source-sink distances and concentration gradients all increase assimilate transport efficiency [12]. Potassium (K), using reduced source-sink distances, increases the cross-sectional area of stems and promotes tuberous root enlargement. Noticeably, the contents of sugars, amino nitrogen (N) compounds and K may be responsible for the osmotic concentrations of phloem sap [13].

In China, an important segment of sweetpotato production is used to make preserved sweetpotato bars (Shu Fu, Shu Fu Gan, Shu Gan, Di Gua Gan, in Chinese), a traditional snack of China. This snack is prepared by cooking the flesh with certain amounts of sucrose. In the last year, the China Agriculture Research System (CARS) began to study the agronomic adaptation and processed quality of three sweetpotato varieties: Zheshu13, Zheshu33 and Jishu26. These varieties are characterized by a high starch content, low moisture and an appropriate texture for processing. Zheshu13 and Zheshu33 are widely used in the Zhejiang province for preserved sweetpotato bar production [14]. Jishu26 is widely used in Shandong province for preserved sweetpotato bar production [15]. However, several characteristics of fresh sweetpotato, such as a bigger fruit size, a thin epidermal layer, a high water content, and fleshy tuberous roots, cause it to become easily wounded during harvesting, packaging, storage, and processing, and then vulnerable to browning and pathogen attack, resulting in the loss of eating and processing quality [16]. These sweetpotato varieties belong to the elongated-vine type, which require the control of their growth. Foliar spraying UCZ can control vine overgrowth and promote the expansion of tuberous roots. Exploring UCZ-affected changes of physicochemical properties in tuberous roots is imperative to this industry.

As we known, tuberous roots of sweetpotato remain vigorous after harvest. Farm chemicals may still affect the changes in tuberous roots. There are still no scientific reports that focus on the effects of UCZ spraying on growth quality during storage, such as flesh color, texture properties, decay rate, and so on, of diverse Chinese sweetpotato varieties. The aim of this study was to evaluate the effects of foliar application of UCZ on the physicochemical properties of tuberous roots in three sweetpotato varieties at the storage stage. The results can provide a more comprehensive understanding of UCZ application effect and storage physiology of sweetpotato that potentially assists producers to use UCZ in production and reduce the loss in the storage of sweetpotatoes.

## 2. Materials and Methods

### 2.1. Plant Materials and Experiment Design

Elongated-vine sweetpotato varieties (Z13, Z33 and J26) were used in this study. These varieties were good for preserved sweetpotato bar in China. Their vine was longer than that of other sweetpotato varieties. They were planted in May and harvested in October 2021



at CARS Sweetpotato Experiment Station, Hefei, Anhui, China (117.26° E, 31.86° N). The physical and chemical properties of soil and the experiment design are given in Table 1. and the uniconazole spraying procedure was according to Duan et al. [5]. We designed two uniconazole treatments: no use and 100 mg·L$^{-1}$ UCZ spraying. The 100 mg·L$^{-1}$ UCZ reagent was prepared by mixing 5% commercially available wettable powder (Sichuan Guoguang Pesticide Co., Ltd., Jianyang, Sichuan, China) and distilled water, which was applied to the corresponding experimental fields. All treatments were uniformly applied using a foliar spray at a rate of 450 L of formulated solution per hectare. Moreover, we designed two fertilizing treatments: normal basal fertilizing, which was 450 kg/hm$^2$ NPK-15:15:15 complex fertilizer; rich basal fertilizing, which was 50 kg/hm$^2$ NPK-15:15:15 complex fertilizer; and 225 kg K$_2$SO$_4$. Chlorpyrifos (10%) (Nanjing Huazhou Pesticide Co., Ltd., Nanjing, Jiangsu, China) was used in the field before being cultivated. The trial was planted in three completely randomized blocks (about 50 m$^2$), and the sweetpotato slip planting density was five plants per square meter. Tuberous roots were stored in Zhejiang A & F University after harvesting. They were stored separately according to the different treatments, and 10 kg of sweetpotatoes were randomly selected for each treatment method to be stored separately. The storage temperature of sweetpotato was about 12 °C, the humidity was about 85%, the storage conditions were unchanged, and samples were taken at 0 DAS, 15 DAS, 30 DAS, 45 DAS and 60 DAS storage, respectively. The samples were completely tuberous roots. They were washed and dried, then cut into small cubes, mixed in a ziplock plastic bag, and placed in the ultra-low refrigerator at −80 °C.

**Table 1.** Uniconazole spraying and fertilization treatment design.

| No. | Uniconazole Treatment | Fertilization Treatment | Soil Properties before Experiment |
|---|---|---|---|
| K1 | No use | Normal basal fertilizing (450 kg·hm$^{-2}$ NPK-15:15:15 complex fertilizer) | Black soil, pH: 5.6; Total N (g·kg$^{-1}$): 1.14; Total P (g·kg$^{-1}$): 0.31; Available N (mg·kg$^{-1}$): 81.91; Available P (mg·kg$^{-1}$): 38.08; Available K (mg·kg$^{-1}$): 160.00; Organic matter (g·kg$^{-1}$): 18.86. |
| K2 | 100 mg·L$^{-1}$ foliar spraying | Normal basal fertilizing | |
| K3 | No use | Rich basal fertilizing (750 kg·hm$^{-2}$ NPK-15:15:15 complex fertilizer and 225 kg K$_2$SO$_4$) | |
| K4 | 100 mg·L$^{-1}$ foliar spraying | Rich basal fertilizing | |

*2.2. Weight Loss, Root Dry Matter and Decay Rate*

Percentage of weight loss was measured according to Lewthwaite et al. [17]. Weight loss rate (%) = (pre-storage mass − post-storage mass)/pre-storage mass × 100%. Root dry matter was measured by the oven-drying method according to van Oirschot et al. [18]. Root dry matter were assessed using approximately 20 g of diced or sliced tissue, dried in an oven at 80 °C for 48 h. Root dry matter rate (%) = mass after drying/mass before drying × 100%. Decay rate was measured according to Yang et al. [19]. Decay incidence was measured during storage. Three replicates of 30 tuberous roots in each treatment were used for evaluation of naturally infected decay during storage. When a spot of naturally growing molds on a root surface was more than 5 mm in width, the root was scored as infected. The proportion of decay was expressed as the percentage of the rotten root to the total.

*2.3. Flesh Color and Brown Index*

The flesh color of sweetpotato was measured according to the methods by Lin [20]. The flesh color of sweetpotato was determined using a chroma meter CR 400 (Konica Minolta Sensing Inc., Sakai, Japan), based on the Commission International de l'Eclairage (CIE) and expressed as the lightness (L*), a* and b* values. From these values, chroma (C) was calculated according to the formula C = $(a^2 + b^2)^{0.5}$. The degree of browning was determined by the extinction value method according to the method described by Liu et al. [21].

### 2.4. Malondialdehyde Content (MDA)

Malonaldehyde (MDA) contents were measured according to Kang et al [22], MDA assay kits (Comin Biotechnology Co., Ltd., Suzhou, China) were used. Triplicate measurements were taken for each sample.

### 2.5. Reducing Sugar Content

The reducing sugar contents were measured according to Pang et al. [23]. Each frozen sample (5.0 g) was homogenized and extracted for 30 min with 50 mL of ethanol, followed by centrifugation at $14,000 \times g$ for 15 min. The supernatant (5.0 mL) was diluted to 50 mL with distilled water, and a mixture containing 3.0 mL of the solution, 2.0 mL of $27.6$ mmol·L$^{-1}$ and 3,5-dinitrosalicylic acid was boiled for 5 min. The reducing sugar content was measured at 520 nm.

### 2.6. Antioxidant Activities

Antioxidant activities were measured by a 2,2-diphenyl-2-picrylhydrazyl (DPPH) radical-scavenging assay according to the method described by Nakagawa [24]. Fifty microliters of 20% ethanol solution and 50 μL of a 200 mM 2-morpholinoethanesulphonic acid buffer (pH 6.0) were added to 50 μL of the sample solution and placed in the wells of a 96-well microplate. The sample solution was prepared by diluting the tuber extract with 100% ethanol solution. The reaction was initiated by the addition of 50 μL of 800 μM DPPH in ethanol. After the reaction mixture had been allowed to stand for 20 min at ambient temperature, its absorbance at 520 nm was measured using a Spectramax 190 microplate reader (Molecular Devices, LLC, San Jose, CA, USA) with Trolox (Sigma-Aldrich, St. Louis, MO, USA) as a standard. The antioxidant activity of these extracts is expressed as mmol Trolox TE·100 g$^{-1}$ FW. The measurements of the sample extracts were triplicated.

### 2.7. Texture Properties

Texture properties were measured according to the methods by Dong [25]. Hardness is the maximum strength peak for the first extrusion cycle. It indicates that when the force exerted on the root as the outside world continues to exert a certain amount of pressure across the root of sweetpotato passes the biological yield point, it reflects the resistance of the sample to deformation, and its unit is N. Springiness is the degree to which the sample can recover after the first compression before the second compression, and it is expressed by the ratio of the height of the second compression to that of the first compression in mm. Chewiness is used to describe the characteristics of solid test samples, defined as the force required by teeth to chew sweetpotato root into the swallowing state.

### 2.8. Statistical Analysis

Statistical analysis was performed using the SPSS software by one-way ANOVA and Duncan's multiple range tests using SPSS 23.0 (IBM, Armonk, NY, USA). Tables were built using Excel 2018 (Microsoft. Co., Redmond, WA, USA), and figures were created using Origin 9.0 Professional (Origin Lab. Co., Northampton, MA, USA) [26]. Statistical analysis was performed using the SPSS software by one-way ANOVA and Duncan's multiple range tests using SPSS 23.0 (IBM, USA). Tables were built using Excel 2018 (Microsoft. Co., USA) and figures were created using Origin 9.0 Professional (Origin Lab. Co., USA) [26].

## 3. Results

### 3.1. Yield of Tuberous Root after Harvest

The yield of the tuberous root is shown in Figure 1. The yield (fresh weight) of the tuberous root showed no significant difference between K1 with K2, and K3 with K4 in all varieties ($p > 0.05$). We found a significant difference between K1 with K4 in Z13 ($p < 0.05$), but Z33 and J26 did not show a significant difference ($p > 0.05$). This indicates that 100 mg·L$^{-1}$ UCZ and high fertilizer maybe increase the yield (fresh weight) of partial varieties.

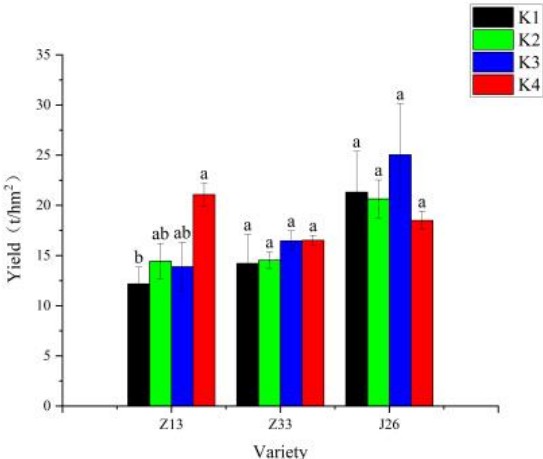

**Figure 1.** Yield of the tuberous root of different UCZ treatments after harvest. Values within the same storage day followed by the same letter are not significantly different at the 5% level according to ANOVA-Duncan's multiple range test. The bar represents standard deviation. K1: Normal basal fertilizing without UCZ; K2: Normal basal fertilizing with UCZ; K3: Rich basal fertilizing without UCZ; K4: Rich basal fertilizing with UCZ.

### 3.2. Decay Rate

Decay rate was a primary indicator in tuberous root storage [27]. We monitored the change in decay rate of tuberous root over 60 days of storage. As shown in Figure 2., the decay rate of all varieties remained at 0.0% until 30 DAS. After 30 DAS, the decay rate of sweetpotato after K1, K3 and K4 treatments increased during storage. The decay rate of K2 was the lowest at any storage time. The decay rate of all the varieties was 0.0% before 45 DAS. Only Z33 increased to 4.4% at 60 DAS. In contrast, the decay rate of K3 was the highest after 30 DAS. K2 and K4 were UCZ spraying treatments, but K4 was higher than K2 in Z13 and J25 at 60 DAS. Foliar application of UCZ during sweetpotato growth may reduce the decay rate during storage. Rich basal fertilizing treatment may increase the decay rate during storage.

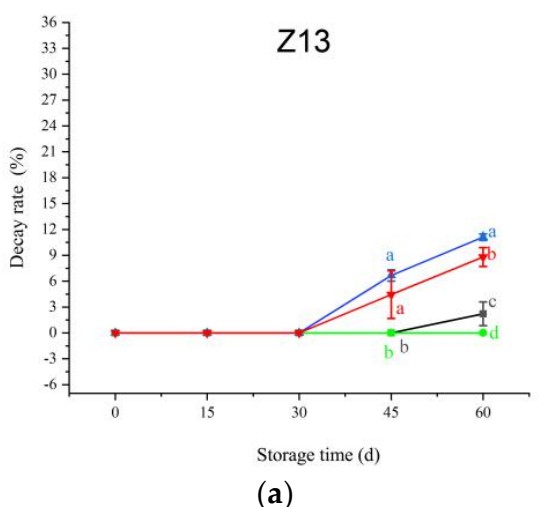

(**a**)

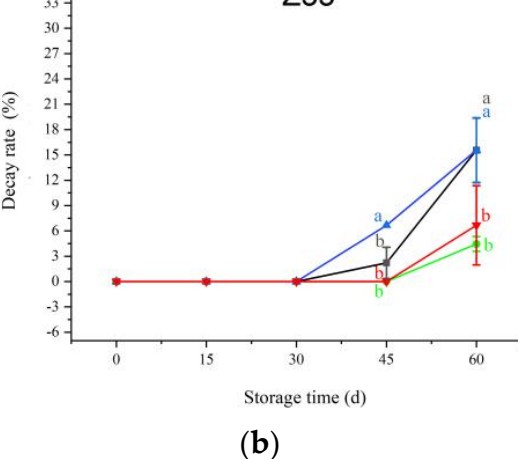

(**b**)

**Figure 2.** *Cont.*

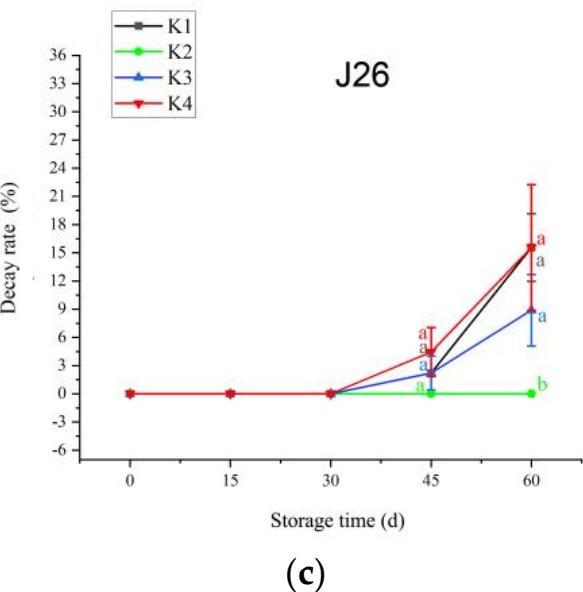

(**c**)

**Figure 2.** Decay rate of the tuberous root of different UCZ treatments during storage. (**a**) Z13; (**b**) Z33; (**c**) J26. Values within the same storage day followed by the same letter are not significantly different at the 5% level according to ANOVA-Duncan's multiple range test. The bar represents standard deviation. K1: Normal basal fertilizing without UCZ; K2: Normal basal fertilizing with UCZ; K3: Rich basal fertilizing without UCZ; K4: Rich basal fertilizing with UCZ.

### 3.3. Dry Matter Rate

Dry matter is an important indicator in evaluating the quality of tuberous roots. We monitored the change in the dry matter rate of tuberous roots after different UCZ treatments under different basal fertilizing levels during storage. Figure 3. shows that all sweetpotato varieties ($p < 0.05$) differed in dry matter rate at DAS 0. The dry matter rate of Z13 was the highest, at 38.02–39.96% (Figure 3a), and J26 was the lowest, at 24.92–27.64% (Figure 3c). The dry matter rate of normal basal fertilizing (K1 and K2) was higher than rich basal fertilizing (K3 and K4) at DAS 0 in Z13 and J26, but all treatments showed insignificant differences at DAS 0 in Z33 ($p > 0.05$). The dry matter rate of K2 and K4 was still higher than K1 during 15–60 DAS in Z13 and J26. Overall, the dry matter rate of tuberous roots with UCZ foliar spraying was higher than with-out UCZ use at 30–60 DAS ($p < 0.05$).

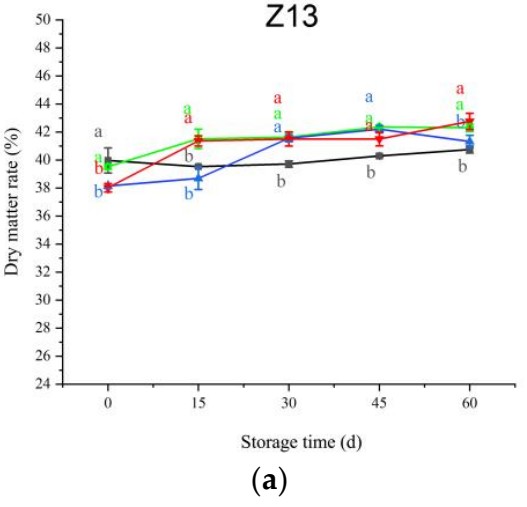

(**a**)

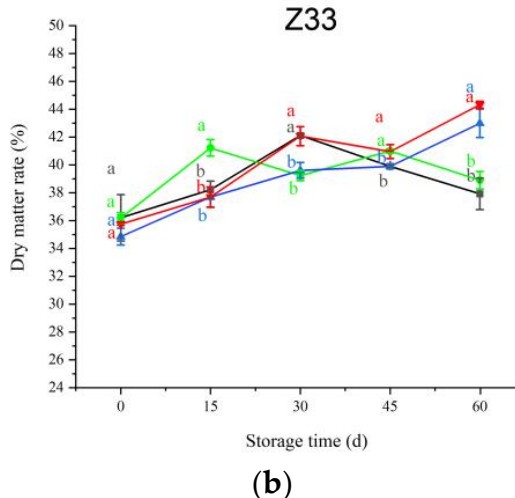

(**b**)

**Figure 3.** *Cont.*

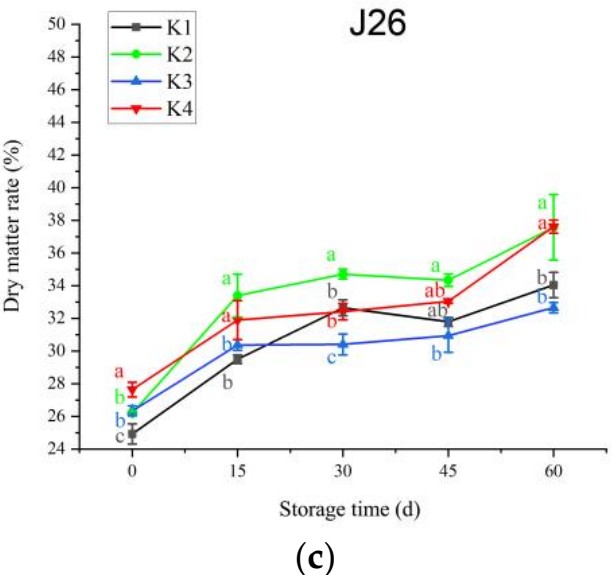

(**c**)

**Figure 3.** Dry matter rate of the tuberous root of different UCZ treatments during storage. (**a**) Z13; (**b**) Z33; (**c**) J26. Values within the same storage day followed by the same letter are not significantly different at the 5% level according to ANOVA-Duncan's multiple range test. K1: Normal basal fertilizing without UCZ; K2: Normal basal fertilizing with UCZ; K3: Rich basal fertilizing without UCZ; K4: Rich basal fertilizing with UCZ.

### 3.4. Weight Loss Rate

According to the weight loss rate of sweetpotatoes, we can determine the extent of internal moisture and other material loss during storage. Weight loss rates are presented in Figure 4. The weight loss rate of the tuberous root increased during storage. The weight loss rates in K2, K3 and K4 treatments were lower than in K1, especially at 60 DAS.

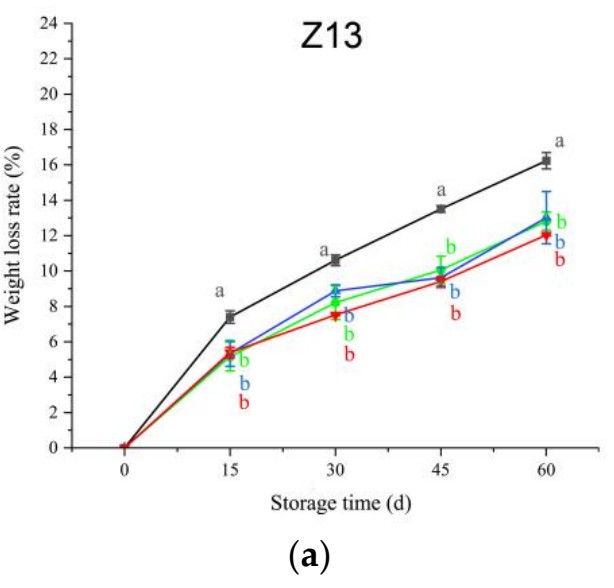

(**a**)

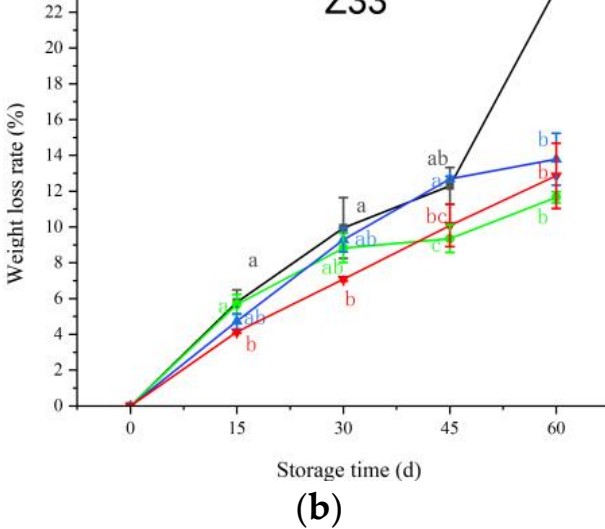

(**b**)

**Figure 4.** *Cont*.

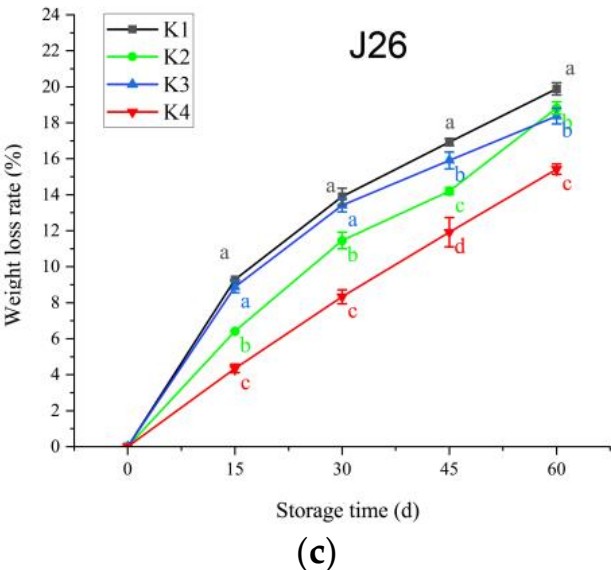

(**c**)

**Figure 4.** Weight loss rate of the tuberous root of different UCZ treatments during storage. (**a**) Z13; (**b**) Z33; (**c**) J26. Values within the same storage day of the same year followed by the same letter are not significantly different at the 5% level according to ANOVA-Duncan's multiple range test. K1: Normal basal fertilizing without UCZ; K2: Normal basal fertilizing with UCZ; K3: Rich basal fertilizing without UCZ; K4: Rich basal fertilizing with UCZ.

### 3.5. Flesh Color and Brown Index

For many tuber and root crops, such as potatoes, yams and sweetpotatoes, the color of the flesh is one of the most essential criteria for the appearance quality of tubers and root crops. In this study, the flesh of both sweetpotato varieties was yellow. As shown in Figure 5a–c, the L* value of flesh decreased during storage in all varieties. In particular, the L* value decreased sharply at 45–60 DAS. The L* value was 73.0–75.0 before storage in all varieties. At 60 DAS, three varieties had different ranges. The L* value of Z13 was 68.0–72.0, that of Z33 was 69.0–73.0, and that of J26 was 68.0–69.0. This indicates that the L* value amplitude of variation of Z13 and Z33 was less than J26. The L* value of rich basal fertilizing treatments (K3 and K4) was still higher than that of normal treatments (K1 and K2) at 15–45 DAS for Z33. Interestingly, UCZ treatments were higher than K1 in all varieties at 60 DAS; however, K2 was sometimes lower than K1, such as at DAS 30 for Z13.

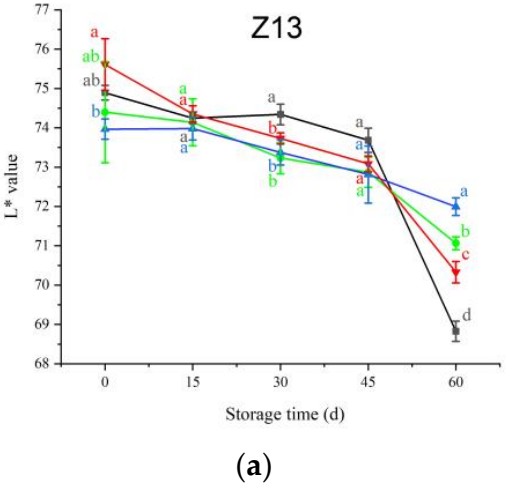

(**a**)

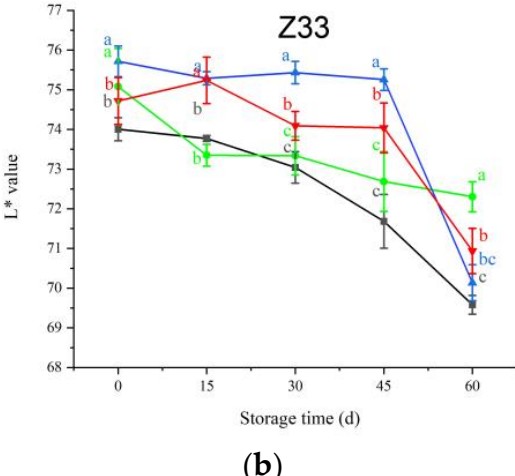

(**b**)

**Figure 5.** *Cont.*

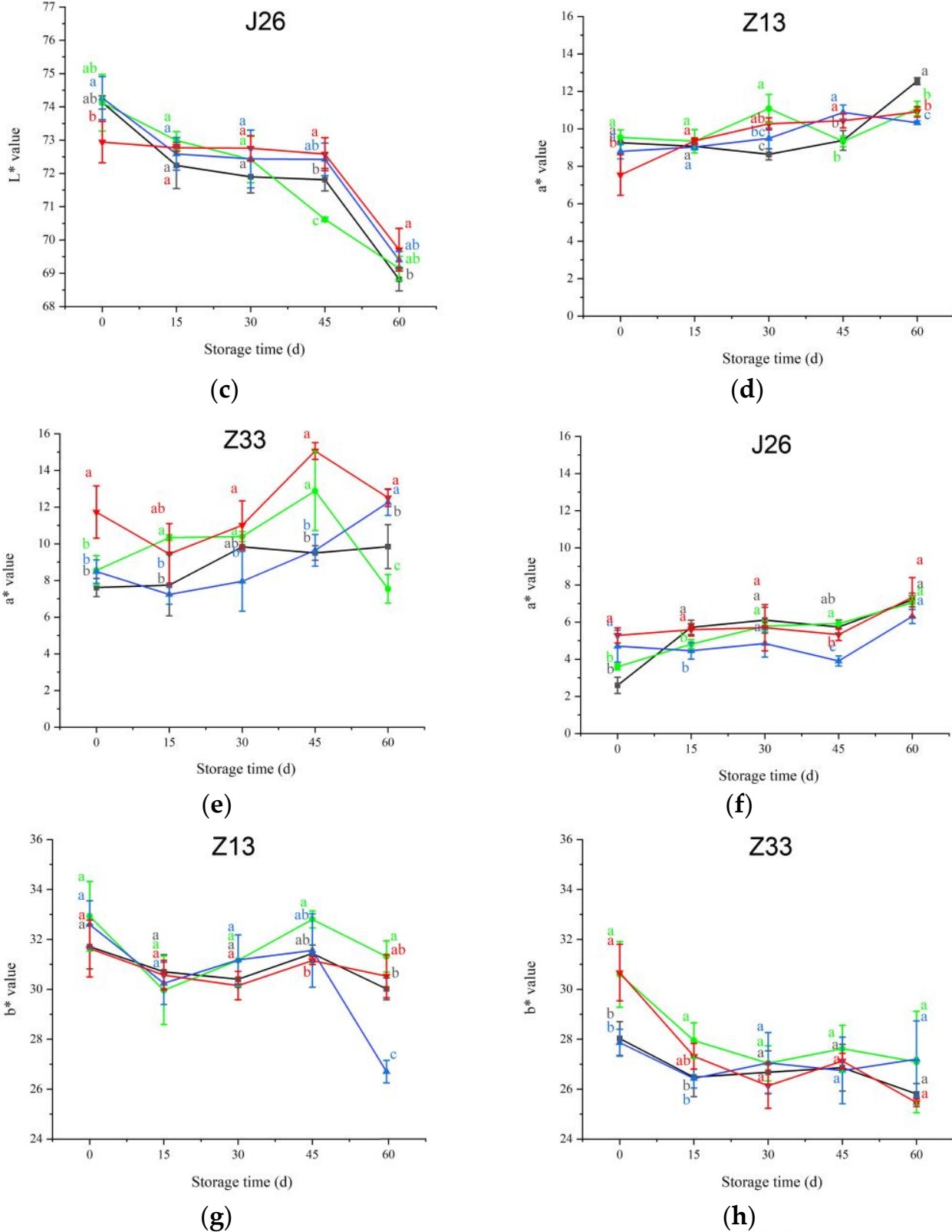

**Figure 5.** *Cont.*

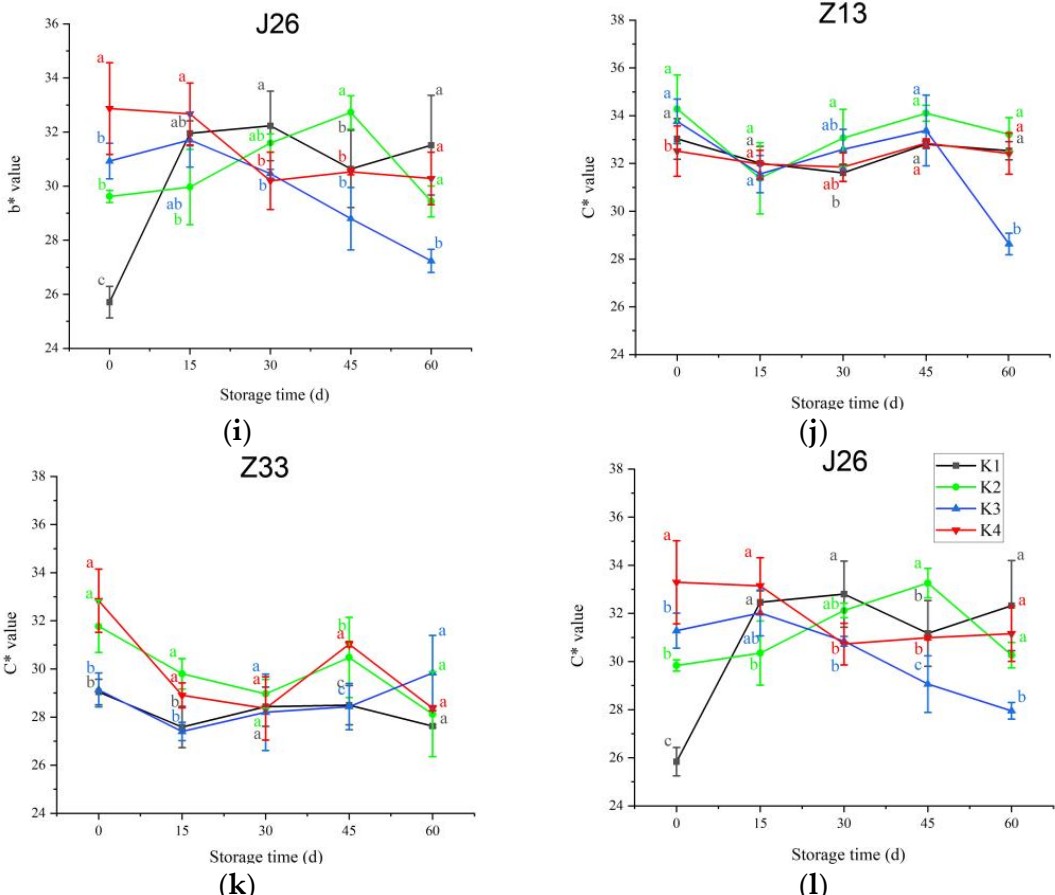

**Figure 5.** Flesh color of the tuberous root of different UCZ treatments during storage. (**a**) L* value of Z13; (**b**) L* value of Z33; (**c**) L* value of J26; (**d**) a* value of Z13; (**e**) a* value of Z33; (**f**) a* value of J26; (**g**) b* value of Z13; (**h**) b* value of Z33; (**i**) b* value of J26; (**j**) C* value of Z13; (**k**) C* value of Z33; (**l**) C* value of J26. Values within the same storage day of the same year followed by the same letter are not significantly different at the 5% level according to ANOVA-Duncan's multiple range test. K1: Normal basal fertilizing without UCZ; K2: Normal basal fertilizing with UCZ; K3: Rich basal fertilizing without UCZ; K4: Rich basal fertilizing with UCZ.

The a* value was always positive in this study, indicating that all varieties were not immature tuberous roots. At 0 DAS, the a* value of Z13 was 7.5–9.5, that of Z33 was 7.5–11.5 and that of J26 was 2.5–5.5. This indicates that Z13 and Z33 were redder than J26. Meanwhile, the a* value of 60 DAS was higher than that of 0 DAS, indicating that the flesh of the tuberous root was browning during storage. The change in the a* value of K2 and K4 was the least. The b* value was positive in this study, decreasing during storage in most cases of the three varieties. It showed an initial decrease and a subsequent increase in Z13 and Z33, then a decrease at 60 DAS. There was a decrease in the C* value for Z13 and Z33, which reflects changes in the flesh color from yellow to dark brown. However, the b* and C* values of K1 were increased at DAS 15–30 in J26. K2 increased at DAS 30–45 in J26. These trends were different from K1 and K2 in other varieties, indicating that reasonable storage may increase the b* and C* values of flesh color in J26.

As shown in Figure 6, Browning is one of the most important reactions taking place during food processing and storage [28]. We monitored tuberous root flesh and the brown index increased between 0 and 60 DAS. It grew fast at 45–60 DAS. The brown index of K3 and K4 had similar values at 0 DAS, indicating that UCZ spraying did not affect the brown index significantly in rich basal fertilizing after harvest ($p < 0.05$). However, the brown index of K2 was lower than K1 at 60 DAS in all varieties. In particular, the brown index of K2 was the lowest at any storage time in Z13. This indicated that UCZ spraying may

inhibit the increase in the brown index during storage in normal basal fertilizing for certain varieties.

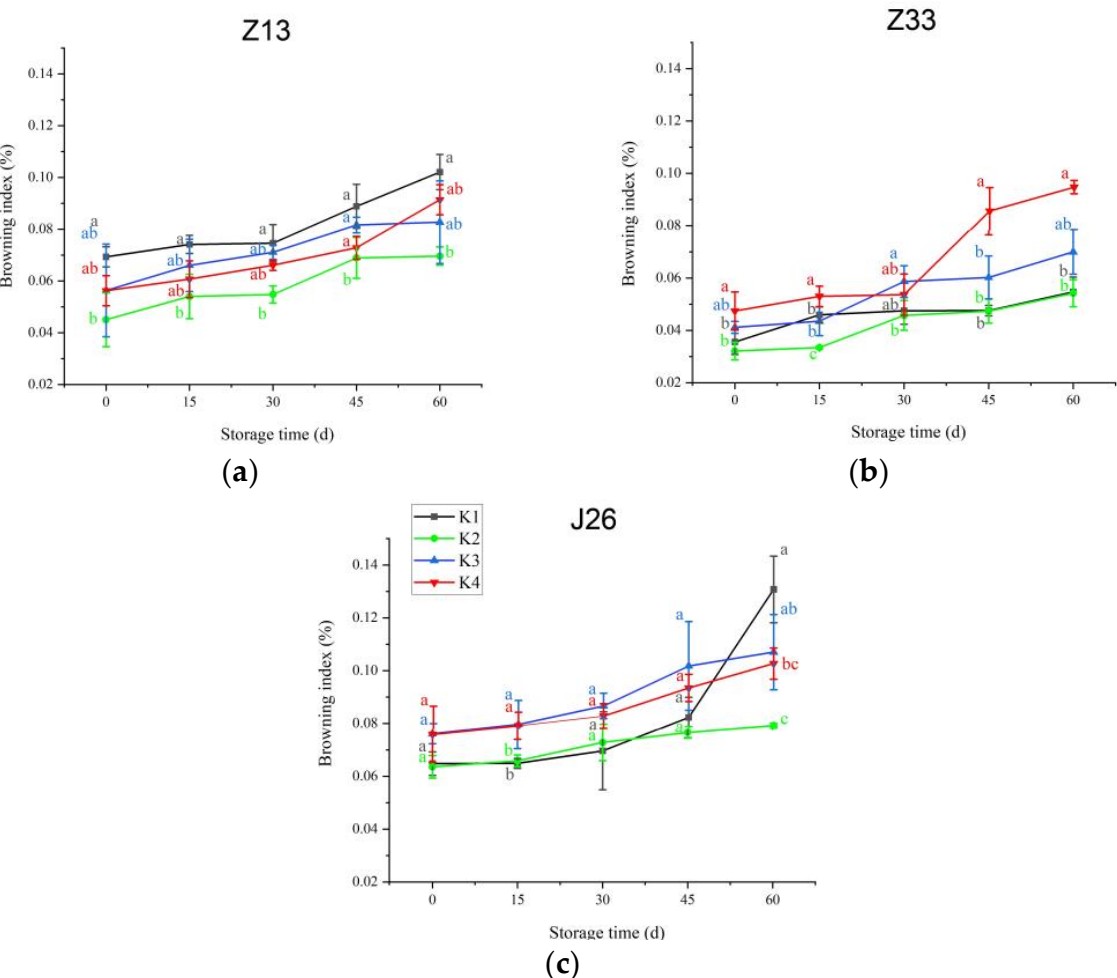

**Figure 6.** Browning index of the tuberous root of different UCZ treatments during storage. (**a**) Browning index of Z13; (**b**) browning index of Z33; (**c**) browning index of J26. Values within the same storage day of the same year followed by the same letter are not significantly different at the 5% level according to ANOVA-Duncan's multiple range test. K1: Normal basal fertilizing without UCZ; K2: Normal basal fertilizing with UCZ; K3: Rich basal fertilizing without UCZ; K4: Rich basal fertilizing with UCZ.

*3.6. Reducing Sugar Content*

Reducing sugars readily interact with amino acids and give rise to Maillard reaction products, which lead to progressive browning [29]. As shown in Figure 7, the reducing sugar content of three sweetpotato varieties showed different changes after different UCZ and basal fertilizing treatments. The reducing sugar content of J26 was lower than that of other varieties at 0 DAS. The reducing sugar content of tuberous root was decreased at 0–15 DAS, and then it was increased at 15–30 DAS in all varieties. The reducing sugar content of Z13 and Z33 was still increased until 60 DAS. However, J26 exhibited a special case, whereby it decreased at 30–45 DAS, then increased at 45–60 DAS. Meanwhile, the reducing sugar content of UCZ spraying treatments (K2 and K4) remained more stable compared to no spraying treatments (K1 and K3) in all varieties. This indicated that UCZ spraying may decrease the reducing sugar content.

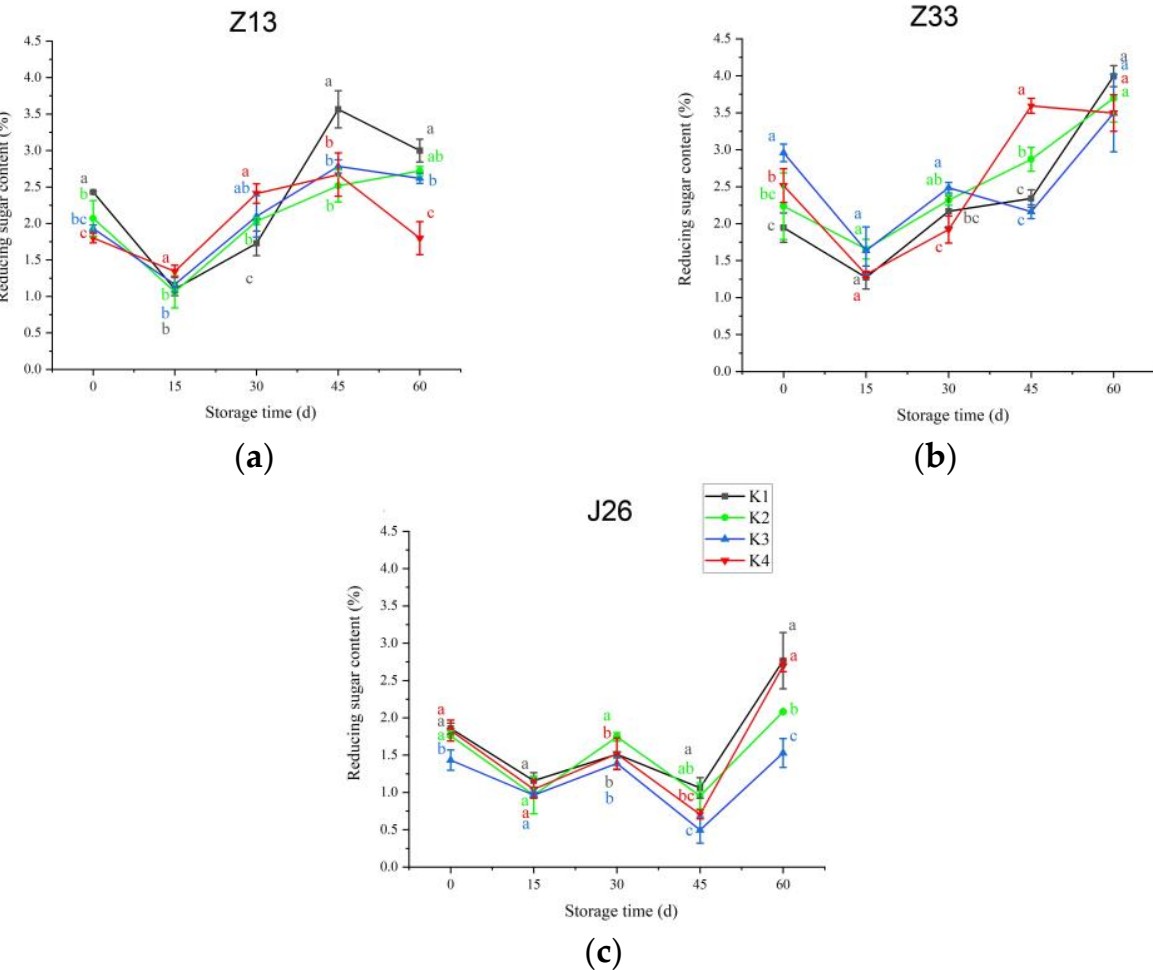

**Figure 7.** Reducing sugar content of different UCZ treatments during storage. (**a**) Z13; (**b**) Z33; (**c**) J26. Values within the same storage day followed by the same letter are not significantly different at the 5% level according to ANOVA-Duncan's multiple range test. K1: Normal basal fertilizing without UCZ; K2: Normal basal fertilizing with UCZ; K3: Rich basal fertilizing without UCZ; K4: Rich basal fertilizing with UCZ.

### 3.7. Malondialdehyde (MDA) Content

As shown in Figure 8, the MDA content of tuberous root was increased during storage in all varieties. It rose slowly at 0–15 DAS, then increased rapidly at 15–45 DAS. In Z13, the MDA content of UCZ spraying treatments (K2 and K4) did not show a noticeable difference compared to no spraying treatments at 0, 45 and 60 DAS, but was higher than K3 at 15 DAS and higher than K1 at 30 DAS. Interestingly, the MDA content of K2 was lower than that of other treatments at any storage time in Z33 and J26. However, the MDA content of K4 was not obviously different for K1 and K3, though it was sometimes higher than for other varieties (at 45 DAS in J26). Although we know that the MDA content of K2 had a low value in this study, it is difficult to conclude that UCZ spraying treatments affected MDA content during storage in this study.

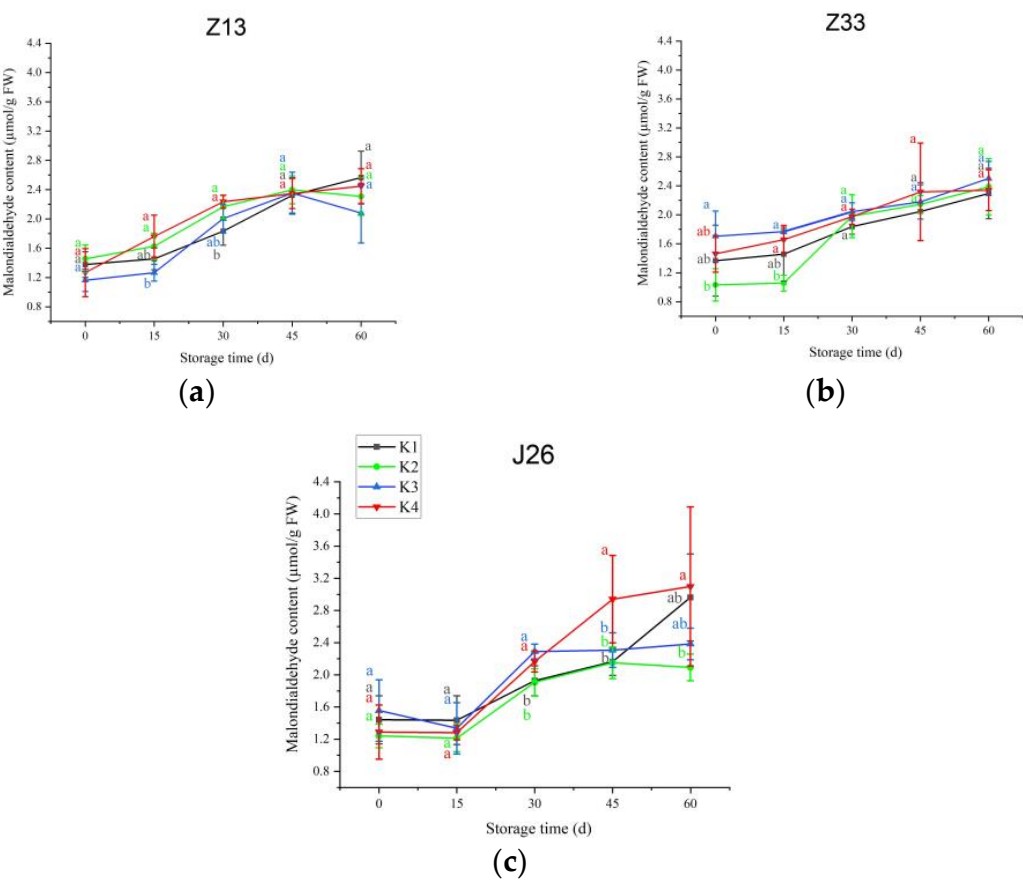

**Figure 8.** Malondialdehyde (MDA) content of the tuberous root of different UCZ treatments during storage. (**a**) Z13; (**b**) Z33; (**c**) J26. Values within the same storage day followed by the same letter are not significantly different at the 5% level according to ANOVA-Duncan's multiple range test. K1: Normal basal fertilizing without UCZ; K2: Normal basal fertilizing with UCZ; K3: Rich basal fertilizing without UCZ; K4: Rich basal fertilizing with UCZ.

### 3.8. Texture Properties

Texture properties are a series of comprehensive concepts that are a good for the quick detection of sweetpotato quality [30]. The hardness of tuberous root is an important textural property. Table 2 shows the hardness of the slice of tuberous root for all varieties. The hardness values of the tuberous root significantly differed between varieties at 0 DAS ($p < 0.05$). The varieties could be ranked, from lowest to highest hardness, as follows: J26 < Z33 < Z13 at 0 DAS. Meanwhile, we found that the changes in hardness for each variety were distinct during storage. The hardness of K1 was significantly harder than that of other treatments at 0 DAS ($p < 0.05$), and K2 was significantly harder than K1 at 30 DAS ($p < 0.05$). At other storage times, the hardness of each treatment showed no significant difference in Z13. In Z33, we found that the hardness increased sharply at 15 DAS and remained stable until 60 DAS for all treatments, except for K4. The hardness of K4 was significantly lower than K3 during storage, except for 30 and 60 DAS ($p < 0.05$). In J26, although the hardness of normal basal fertilizing treatments (K1 and K2) was lower than rich fertilizing treatments (K3 and K4) at 0 DAS, the hardness of K1 and K2 was increased during storage, where K1 was increased sharply in particular. Thus, it is difficult to suggest that UCZ spraying directly affects the hardness of tuberous root in sweetpotato.

**Table 2.** Hardness of the slice of tuberous root (N).

| Variety | DAS | Treatment | | | |
|---|---|---|---|---|---|
| | | **K1** | **K2** | **K3** | **K4** |
| Z13 | 0 | 138.3 ± 4.0 [ABa] | 132.1 ± 2.5 [ABb] | 127.7 ± 3.5 [Bbc] | 121.8 ± 2.8 [Cc] |
| | 15 | 142.8 ± 3.6 [Aa] | 137.5 ± 5.0 [Aa] | 135.8 ± 7.5 [ABa] | 146.3 ± 5.3 [Aa] |
| | 30 | 126.9 ± 3.0 [Cab] | 129.9 ± 2.0 [Ba] | 127.1 ± 6.7 [Bab] | 119.6 ± 1.1 [Cb] |
| | 45 | 139.5 ± 3.0 [ABa] | 139.0 ± 6.0 [Aa] | 142.1 ± 1.4 [Aa] | 136.4 ± 1.3 [Ba] |
| | 60 | 134.7 ± 5.7 [Ba] | 137.1 ± 0.7 [Aa] | 139.31 ± 1.9 [Aa] | 134.8 ± 3.5 [Ba] |
| Z33 | 0 | 116.6 ± 2.1 [Ba] | 121.9 ± 3.3 [Ba] | 119.8 ± 3.6 [Ba] | 108.8 ± 2.2 [Bb] |
| | 15 | 132.3 ± 0.9 [Aab] | 137.4 ± 2.9 [Aa] | 125.6 ± 5.9 [ABb] | 115.0 ± 4.4 [Bc] |
| | 30 | 132.3 ± 7.2 [Aa] | 134.8 ± 9.5 [Aa] | 126.1 ± 7.3 [ABa] | 120.0 ± 6.2 [Ba] |
| | 45 | 133.1 ± 3.9 [Ab] | 141.6 ± 2.4 [Aa] | 134.9 ± 0.7 [Aab] | 132.9 ± 5.8 [Ab] |
| | 60 | 132.9 ± 0.1 [Aab] | 125.2 ± 1.9 [Bb] | 125.9 ± 7.2 [ABb] | 139.8 ± 9.8 [Aa] |
| J26 | 0 | 71.3 ± 2.7 [Cc] | 80.6 ± 1.0 [Bb] | 90.1 ± 6.2 [ABa] | 96.4 ± 1.2 [ABa] |
| | 15 | 94.0 ± 0.8 [Bab] | 89.2 ± 1.2 [Ab] | 92.8 ± 3.5 [Aab] | 97.8 ± 3.9 [Aa] |
| | 30 | 94.5 ± 2.6 [Ba] | 90.5 ± 4.2 [Aa] | 82.8 ± 4.2 [Bb] | 94.8 ± 4.3 [ABa] |
| | 45 | 92.7 ± 4.4 [Ba] | 91.6 ± 4.8 [Aa] | 88.6 ± 6.6 [ABa] | 90.8 ± 1.3 [Ba] |
| | 60 | 103.8 ± 2.4 [Aa] | 91.0 ± 0.9 [Ab] | 91.2 ± 1.4 [ABb] | 85.4 ± 1.8 [Cc] |

Note: Different capital letters indicate a significant difference at the 5% level according to ANOVA-Duncan's multiple range test between different storage days; different lowercase letters indicate a significant difference at the 5% level according to ANOVA-Duncan's multiple range test between different treatments. K1: Normal basal fertilizing without UCZ; K2: Normal basal fertilizing with UCZ; K3: Rich basal fertilizing without UCZ; K4: Rich basal fertilizing with UCZ.

In Table 3, the springiness range of the tuberous root of three varieties was 4.4 to 5.4 at 0 DAS. The changing springiness of the slice of tuberous root was different among the three varieties during storage. The springiness of the slice of tuberous root of K1, K2 and K3 was significantly increased at 30 DAS in Z13, then it decreased at 45 or 60 DAS ($p < 0.05$). The springiness of K4 did not change sharply during storage ($p > 0.05$), and was significantly higher than that of K1 and K3 in Z13 at 0 DAS ($p < 0.05$), and no significant difference in springiness was observed between all treatments at other DAS in Z13. In Z33 ($p > 0.05$), the springiness of the slice of tuberous root of K2 was significantly increased at 15 DAS ($p < 0.05$), and K1 and K3 at 30 DAS ($p < 0.05$), and K4 was sharply decreased at 15 DAS then increased at 60 DAS. The springiness of K3 was significantly higher than that of other treatments at 0 DAS, except K1 ($p > 0.05$). In J26, the highest springiness was always found at 30 DAS. The springiness of each treatment did not show a significant difference at 45 and 60 DAS. The springiness of K1 was the highest at 30 DAS, and that of K4 was the highest at 15 DAS. In brief, the springiness was the highest at 15–30 DAS in both varieties. Thus, there is insufficient evidence to conclude that UCZ affects the springiness of tuberous roots.

Table 4 shows that the chewiness values of tuberous root slices were significantly different between varieties at 0 DAS ($p < 0.05$). The varieties could be ranked at 0 DAS, from lowest to highest chewiness, as follows: J26 < Z33 < Z13. The chewiness of K2 was not significantly different from that of K1 at 0 DAS in all varieties ($p > 0.05$), and the chewiness of K4 was not significantly different from that of K3 in Z13 and Z33 ($p > 0.05$); only K4 was significantly lower than K3 in J26 ($p < 0.05$). In Z13, the chewiness of K1 was not significantly different from that of K2 at all storage times, except 30 DAS ($p > 0.05$). At 30 DAS, the chewiness of K2 was higher than that of K1, and that of K3 was higher than K4 at 15 DAS but lower than K4 at 30 DAS. In Z33, the chewiness of K1 was insignificantly different from K2 at all storage times except for 45 DAS ($p > 0.05$). The chewiness of K2 was significantly higher than that of K1 at 45 DAS ($p < 0.05$), and that of K4 was significantly lower than K3 at 30 and 45 DAS ($p < 0.05$). In J26, only the chewiness of K2 was significantly lower than that of K1 at 15 DAS ($p < 0.05$). This showed that UCZ spraying did not change the chewiness values of tuberous roots obviously in most cases before storage. UCZ may affect the chewiness values during storage time, but no obvious trends were found in this study. It was hard to conclude that UCZ spraying affects the springiness values directly.

**Table 3.** Springiness of the slice of tuberous root (N).

| Variety | DAS | Treatment | | | |
|---|---|---|---|---|---|
| | | **K1** | **K2** | **K3** | **K4** |
| Z13 | 0 | 4.8 ± 0.2 ABb | 5.0 ± 0.2 Bab | 4.8 ± 0.3 Bb | 5.4 ± 0.3 Aa |
| | 15 | 5.0 ± 0.3 Aa | 5.0 ± 0.3 Ba | 4.8 ± 0.1 Ba | 5.2 ± 0.3 Aa |
| | 30 | 5.1 ± 0.2 Aa | 5.5 ± 0.1 Aa | 5.3 ± 0.3 Aa | 5.3 ± 0.4 Aa |
| | 45 | 4.6 ± 0.1 Ba | 4.8 ± 0.2 Ba | 4.9 ± 0.4 ABa | 4.9 ± 0.2 Aa |
| | 60 | 4.6 ± 0.1 Ba | 4.9 ± 0.3 Ba | 4.6 ± 0.1 Ba | 4.9 ± 0.3 Aa |
| Z33 | 0 | 5.0 ± 0.2 Bab | 4.4 ± 0.4 Bc | 5.4 ± 0.2 Aa | 4.8 ± 0.1 Abc |
| | 15 | 5.0 ± 0.2 BCab | 5.1 ± 0.1 Aa | 4.8 ± 0.2 Bb | 4.4 ± 0.1 Cc |
| | 30 | 5.6 ± 0.1 Aa | 4.8 ± 0.0 ABb | 5.5 ± 0.2 Aa | 4.6 ± 0.0 Bb |
| | 45 | 4.9 ± 0.1 BCab | 5.2 ± 0.4 Aa | 5.0 ± 0.2 Bab | 4.5 ± 0.2 BCb |
| | 60 | 4.6 ± 0.3 Cbc | 5.0 ± 0.1 Aab | 4.4 ± 0.2 Cc | 5.1 ± 0.2 Aa |
| J26 | 0 | 5.1 ± 0.0 Ba | 4.9 ± 0.4 Aa | 5.1 ± 0.2 ABa | 4.4 ± 0.2 Bb |
| | 15 | 5.3 ± 0.2 Bab | 5.0 ± 0.2 Ab | 5.0 ± 0.3 ABCb | 5.5 ± 0.2 Aa |
| | 30 | 5.7 ± 0.1 Aa | 5.1 ± 0.1 Ab | 5.2 ± 0.5 Ab | 4.9 ± 0.2 ABb |
| | 45 | 4.9 ± 0.5 Ba | 4.5 ± 0.2 Aa | 4.4 ± 0.3 Ca | 4.9 ± 0.4 ABa |
| | 60 | 5.2 ± 0.3 Ba | 4.6 ± 0.8 Aa | 4.6 ± 0.1 BCa | 5.0 ± 0.5 ABa |

Note: Different capital letters indicate a significant difference at the 5% level according to ANOVA-Duncan's multiple range test between different storage day; different lowercase letters indicate a significant difference at the 5% level according to ANOVA-Duncan's multiple range test between different treatments at same storage day. K1: Normal basal fertilizing without UCZ; K2: Normal basal fertilizing with UCZ; K3: Rich basal fertilizing without UCZ; K4: Rich basal fertilizing with UCZ.

**Table 4.** Chewiness of the slice of tuberous root (ratio).

| Variety | DAS | Treatment | | | |
|---|---|---|---|---|---|
| | | **K1** | **K2** | **K3** | **K4** |
| Z13 | 0 | 129.9 ± 11.0 Bab | 136.4 ± 8.6 Ba | 114.4 ± 1.6 Cb | 129.5 ± 9.4 BCab |
| | 15 | 153.7 ± 2.3 Aa | 140.8 ± 12.9 ABa | 117.3 ± 7.9 Cb | 153.5 ± 11.0 Aa |
| | 30 | 129.9 ± 0.9 Bb | 153.9 ± 5.4 Aa | 128.0 ± 9.7 Bbc | 116.2 ± 6.1 Cc |
| | 45 | 131.3 ± 14.7 Bb | 134.3 ± 7.9 Bb | 142.3 ± 2.2 Aab | 154.0 ± 9.4 Aa |
| | 60 | 131.5 ± 12.9 Ba | 130.9 ± 8.6 Ba | 130.4 ± 1.8 Ba | 133.0 ± 4.9 Ba |
| Z33 | 0 | 117.6 ± 3.1 Aa | 116.8 ± 9.2 Ca | 123.0 ± 8.7 ABCa | 107.0 ± 19.1 Aa |
| | 15 | 127.6 ± 5.8 Aa | 133.1 ± 1.0 Ba | 103.6 ± 12.3 Cb | 100.8 ± 8.3 Ab |
| | 30 | 125.1 ± 5.5 Aa | 122.0 ± 8.2 Ca | 127.5 ± 6.5 ABa | 102.1 ± 3.7 Ab |
| | 45 | 125.0 ± 8.2 Abc | 148.9 ± 1.9 Aa | 130.9 ± 11.0 Ab | 114.2 ± 8.5 Ac |
| | 60 | 127.1 ± 8.1 Aa | 116.2 ± 3.8 Cab | 109.5 ± 14.2 BCb | 115.8 ± 2.9 Aab |
| J26 | 0 | 82.8 ± 6.6 Bb | 83.7 ± 14.5 Ab | 106.1 ± 10.6 Aa | 86.3 ± 8.2 Ab |
| | 15 | 104.6 ± 1.2 Aa | 81.4 ± 2.1 Ab | 85.4 ± 4.8 Bb | 91.5 ± 13.6 Aab |
| | 30 | 81.0 ± 6.2 Bab | 65.2 ± 4.5 Bb | 67.8 ± 14.0 Cab | 81.5 ± 3.4 ABa |
| | 45 | 67.6 ± 3.7 Ca | 63.2 ± 3.4 Ba | 67.9 ± 1.8 Ca | 68.5 ± 9.4 Ba |
| | 60 | 106.9 ± 9.6 BCa | 75.6 ± 3.2 ABa | 66.7 ± 3.3 Ca | 69.4 ± 2.8 Ba |

Note: Different capital letters indicate a significant difference at the 5% level according to ANOVA-Duncan's multiple range test between different storage day; different lowercase letter indicate a significant difference at the 5% level according to ANOVA-Duncan's multiple range test between different treatments. K1: Normal basal fertilizing without UCZ; K2: Normal basal fertilizing with UCZ; K3: Rich basal fertilizing without UCZ; K4: Rich basal fertilizing with UCZ.

### 3.9. DPPH Radical-Scavenging Activity

As Figure 9 shown, as storage time proceeded, the changes in DPPH radical-scavenging activity were disordered in all varieties, increasing at 15 DAS, then decreased at 30 DAS, and increasing again at 45 DAS. The scavenging activity of practical treatments remained stable at 45–60 DAS, but in some treatments, it decreased at 60 DAS. In the vast majority of cases, DPPH radical-scavenging activity of K2 was lower than that of K1, and that of K4 was higher than K3. Although the DPPH radical-scavenging activity of K2 was always lower than that of other treatments, it is hard to indicate that UCZ spraying treatments

reduce DPPH radical-scavenging activity, because the DPPH radical-scavenging activity of K4 was not in accordance with K3 in most cases.

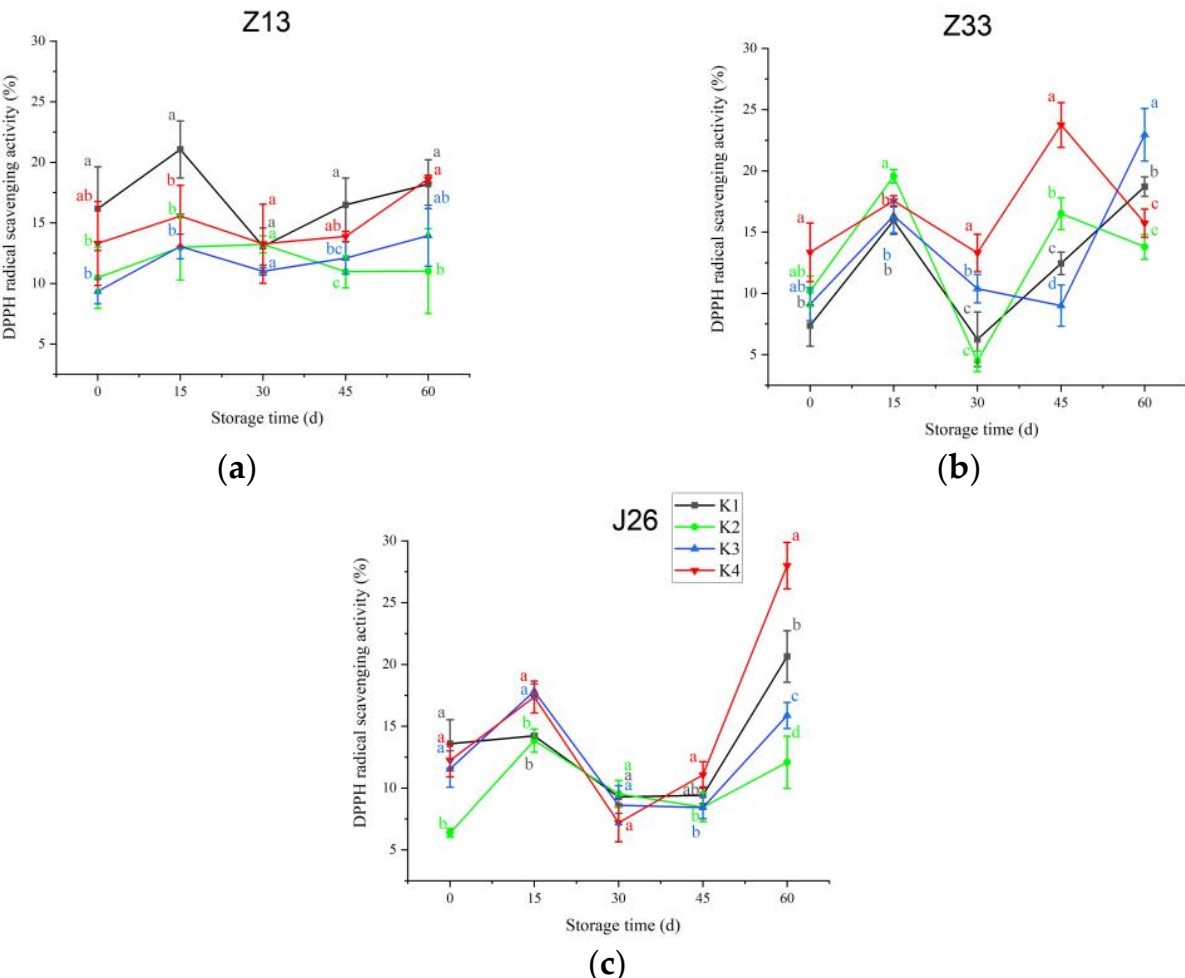

**Figure 9.** DPPH radical-scavenging activity of the tuberous root of different UCZ treatments during storage. (**a**) Z13; (**b**) Z33; (**c**) J26. Values within the same storage day followed by the same letter are not significantly different at the 5% level according to ANOVA-Duncan's multiple range test. K1: Normal basal fertilizing without UCZ; K2: Normal basal fertilizing with UCZ; K3: Rich basal fertilizing without UCZ; K4: Rich basal fertilizing with UCZ.

## 4. Discussion

Foliar UCZ application promotes *C. betacea* seedling growth and improves nutrition accumulation ability [31]. Tuberous root yield, tuberous root weight per plant and the number of tuberous roots per plant was increased obviously after UCZ spraying [5,32]. We found a similar trend in this study, whereby foliar application of UCZ had a positive effect on the storage quality of tuberous roots. It reduced the decay rate, dry matter rate and weight loss rate during the storage time period (Figures 2–4). UCZ has a bacteriostatic effect; plant absorption can improve the surviving rate under growth [33,34]. A previous study reported that the biomass ratios of the leaf and stem are markedly decreased, whereas the biomass ratio of the root is significantly enhanced following foliar application of UCZ [35]. UCZ inhibited gibberellic acid (GA) and promoted abscisic acid (ABA) in flag leaves. A higher ABA content in tuberous roots is beneficial for the extension of storage time, and a lower GA content is good for inhibiting the sprouting of tuberous roots during storage.

Low sucrose levels relative to starch levels were associated with reduced sucrose export rates and a high sucrose–starch ratio in leaves was necessary for sucrose transport from source to sink [36]. High hardness and springiness values of tuberous roots usually

indicate a high starch content. We found that different varieties exhibited a unique situation during storage after foliar UCZ spraying. The hardness of J26 was still increasing during storage. Z33 showed no significant difference during storage.

UCZ application inhibited stem elongation and decreased the average source–sink distance during the mid and late developmental phases of both varieties. The foliar application of UCZ to sweetpotato plants significantly reduces dry matter accumulation in the above-ground parts and accelerates below-ground growth [37]. We confirmed that in J26 from 0 to 60 DAS, the dry matter rates of UCZ spraying treatments were higher than those using no treatment, meanwhile, they were tenable in Z13 from 15 to 60 DAS. However, in Z33, there was no significant difference between treatments that did or did not use UCZ spraying.

In this study, we found that the storage quality of rich basal fertilizing treatments was unstable. Occasionally, the storage quality of rich basal fertilizing treatments was not better than that of normal fertilizing treatments, such as the decay rate of Z13 changing at 60 DAS in this study. This indicated that the fertilizer should be in accordance with the nutrient absorption characteristics of the variety. Variation in the potassium concentration, accumulation and potassium efficiency ratio existed among genotypes in sweetpotato [38]. Thus, it is possible that excessive fertilization may reduce the storage quality of tuberous root. Rich basal fertilizing may slow down the weight loss rate. Foliar application of UCZ during sweetpotato growth may retard the increase in weight loss rate during storage (Figure 4). Reasonable fertilization management with UCZ foliar spraying is a method of adjustment to achieve high storage quality.

Sweetpotatoes of different varieties and cultivation vary in their flesh colors and physicochemical properties [39]. A previous study found that potassium positively affects skin characteristics of tuberous roots in sweetpotato, and supplementing potassium in sandy soils improved sweetpotato yield and contributed to a better organization of the skin of tuberous roots [40]. In this study, we found that the flesh color of rich fertilizing treatments was significantly different compared to that of normal fertilizing treatments. The flesh color of sweetpotato undergoes browning during storage, a process whereby the flesh color darkens, accompanied by an increase in reducing sugar content during storage [41]. In this study, the browning index was significantly different between rich fertilizing treatments and normal fertilizing treatments at 0 to 45 DAS. The browning index of UCZ spraying treatments was not significantly different compared to that of no UCZ application treatments. Texture properties were affected by the cell-binding force [21]. In this study, we found different trends of hardness and chewiness in different varieties. However, there was insufficient evidence to prove that UCZ has an influence on texture quality.

UCZ foliar application may improve the antioxidant system [42,43]. UCZ dry seed dressing reduced MDA content and improved stress resistance during the growth of wheat [44]. We found similar trends in this study, whereby the MDA content of K2 in Z33 was lower than that not using UCZ treatment. UCZ foliar application at 0 and 15 DAS may reduce the MDA content in some sweetpotato varieties. However, the DPPH radical-scavenging activity following UCZ foliar application was lower compared to the no spraying treatment under normal basal fertilizing during tuberous root storage time. This indicates the possibility of UCZ foliar application to reduce DPPH radical-scavenging activity.

Rich foliar potassium fertilization can increase the level of phenolic compounds and the corresponding antioxidant activity of sweetpotato at the growth stage [45]. However, we found that rich basal potassium fertilization did not decrease the loss of tuberous root, and the MDA content did not show a significant difference between rich and normal basal fertilization. Occasionally, DPPH radical-scavenging activity was lower than normal basal potassium fertilization. Perhaps foliar potassium fertilization was better than basal fertilization for the storage quality of tuberous roots.

## 5. Conclusions

The effects of UCZ treatment on sweetpotato are likely very complex. Compared with a control under regular fertilization, foliar UCZ application could increase the flesh color of the tuberous roots. Foliar UCZ applications generated desirable storage to a certain extent. The decay rate and weight loss rate decreased under 100 mg·L$^{-1}$ UCZ spraying with normal basal fertilizing treatment in all varieties. This inhibited the decrease of the brown index to a certain degree and slowed down the decline rate of the L* value. It is worth noting that the variety of genes should be considered when using UCZ. In the present study we know that the use of UCZ did not significantly affect the decrease in storage quality of tuberous roots ($p < 0.05$). It is still necessary to further explore the regulating mechanisms of uniconazole by molecular means in the future to provide more powerful guidance for UCZ application in sweetpotato.

**Author Contributions:** Conceptualization, G.L.; Formal analysis, X.X. and X.P.; Funding acquisition, G.L.; Investigation, X.P. and H.Z.; Methodology, Z.L., L.P. and G.L.; Resources, J.X., P.C. and Y.C.; Supervision, X.X. and G.L.; Visualization, X.X. and X.P.; Writing—original draft, X.X. and X.P.; Writing—review & editing, X.X., M.S.G. and G.L. All authors have read and agreed to the published version of the manuscript.

**Funding:** This research was funded by the Key research and development program of Zhejiang province (2021C02057), China Agriculture Research System (CARS-10-GW21), the Scientific Research Foundation for the introduction of talent by Zhejiang A&F University (2021LFR017), and the Key Laboratory of Marine Fishery Resources Exploitment Exploitation & Utilization of Zhejiang Province (SL2022016).

**Data Availability Statement:** Not applicable.

**Acknowledgments:** Would like to acknowledge the contributions of technical staff Xiafang Ye and Bixin Yao for their technical support of the project. All authors have read and agreed to the published version of the manuscript.

**Conflicts of Interest:** The authors declare no conflict of interest.

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
