# Peer review of "Effects of Foliar Application of Uniconazole on the Storage Quality of Tuberous Roots in Sweetpotato"

_agronomy, doi:10.3390/agronomy12122983_

Round 1
Reviewer 1 Report
The manuscript is very interesting and poses very important information on the use of UCZ and fertilizer in the postharvest quality of Sweetpotato. The document is very well written, it has sufficient theoretical support, the experimental phase is well described and approached, as it is written it allows it to be replicated, and it has the respective statistical support. The results are well described. I suggest improving the quality of the figures, and expanding the discussion chapter including more details about the UCZ in plant physiology and biochemistry.

Author Response
Response to Reviewer 1 Comments
Dear Editor and Reviewers:
On behalf of my co-authors, we are very grateful to you for giving us an opportunity to revise our manuscript. we appreciate you very much for your positive and constructive comments and suggestions on our manuscript entitled “Effects of foliar application of uniconazole on the storage quality of tuberous roots in sweetpotato” (ID: agronomy-2022126).We have studied reviewers’ comments carefully and tried our best to revise our manuscript according to the comments. The following are the responses and revisions I have made in response to the reviewers' questions and suggestions on an item-by-item basis. Thanks again to the hard work of the editor and reviewer!
Point 1: Title: if the fertilization factor is included
Response 1: Thanks for your suggestion. The point is UCZ spraying, if fertilization factor noticed, the topic would be changed. So, we leave the title as it is.
Point 2: Line 33: scientific name in cursive; Line 49: tolerance to? Line 61 Potassium (K) ;Line 63: replace potassium with K; Line 73: …low moisture and appropriate texture for processing (include citation);Line 81: change explore by evaluate.
Response 2: Thank you very much for discovering these errors. We aplilgize for this problems and have corrected it based on your suggestions. L73, we delete this sentence because we found it was not suitable.
Point 3: Materials and methods
Line 93: include details of the reagent used as the source of UCZ, eg concentration? Comercial product? Reagent? Line 94: change basic by normal; Line 96: Statistically, what do the blocks correspond to? Ground slope? Fertility? Line 114: in Decay rate: briefly describe the methodology; Lines 114-115: the reducing sugar content methodology is described in chapter 2.5, leave only one description; Line 132-133: L-1, the -1 in superscript; 2.8. Statistical analysis: before Duncan's new multiple range method, it should be noted that Anova was performed, after completing the normality test and the variance homogeneity test.
Response 3: Thank you very much for discovering these errors. We followed your suggestions to modify the materials and methods, the detail of UCZ was added, basic was changed normal. The detail of soil properties was added in Table.1. Added the decay rate and adjusted reducing sugar content methodology. ANOVA was performed.
Point 4: Results
Figure 1: The figures are very small and do not have good resolution, I suggest improving their quality. Line 188: tuberous root of three varieties (Z13, Z33 and J26). Line 189: I don't understand why they include "the same year"? Indicate what the vertical bars are, for example the standard error or standard deviation. Briefly define K1, K2, K3 and K4, for example: K1: normal fertilizer, not UCZ. These suggestions include them in the other figures
Line 197-198: this information is better in the discussion
Table 2: complete the title, for example Postharvest Hardness (N) of slice of tuberous root with application of potassium and UCZ. I also suggest including a description of the treatments for a better understanding of the table. ± is the standard error? Or the standard deviation? Include this suggestion in the other results tables.
Response 4: Thank you very much for discovering these errors. We followed your suggestions to adjust the figure, we improved their quality. And we added the notes inside. ± is the standard deciation.
Point 5: Discussion
Line 348-352: delete or summarize this part of the discussion, it does not help to explain your results
Line 354: …during storage time. Cite the figure with these results
Discuss the results in the same order they were presented, start with decay rate
Lack of discussion of the variables related to texture (Hardness, Chewiness, Adhesiveness)
The positive effect of UCZ should be explained in more detail, based for example on the fact that it decreases the synthesis of gibberellic acid.
Response 5: We delete this part of the discussion. And added the texture properties discussion. Higher ABA in the tuberous roots is good for the extension of storage time, and lower GA is good for inhibiting the tuberous root sprout during storage.
In general, we think your opinion is constructive. It has been modified according to your suggestions.

Reviewer 2 Report
It is necessary to indicate the coordinates of the research site in the “Materials and Methods” section.
The research conditions are not specified: climatic and soil.
There is no technology of cultivation . There is no plant protection system, which significantly affects storage conditions (specific and quantitative composition of pests and diseases).
It is necessary to indicate whether the varieties were dressed with pesticides or not.
The characteristics of the studied varieties are not indicated: features of development, resistance to adverse environmental conditions, reproduction index, quality, etc.
When describing experimental data, it is necessary to indicate the results of statistical data processing: whether the results are reliable or not.
The conclusions are very general. It is necessary to draw conclusions with concrete results.
Author Response
Dear Editor and Reviewers:
On behalf of my co-authors, we are very grateful to you for giving us an opportunity to revise our manuscript. we appreciate you very much for your positive and constructive comments and suggestions on our manuscript entitled “Effects of foliar application of uniconazole on the storage quality of tuberous roots in sweetpotato” (ID: agronomy-2022126).We have studied reviewers’ comments carefully and tried our best to revise our manuscript according to the comments. The following are the responses and revisions I have made in response to the reviewers' questions and suggestions on an item-by-item basis. Thanks again to the hard work of the editor and reviewer!
Point 1: It is necessary to indicate the coordinates of the research site in the “Materials and Methods” section.
Response 1: Please provide your response for Point 1. (in red)
Point 2:The research conditions are not specified: climatic and soil.
Response 2: Thanks for your suggestion, we add soil peoperties in Table 1..
No. |
Uniconazole treatment |
Fertilization treatment |
Soil properties before experiment |
K1 |
No use |
Normal basal fertilizing (450 kg·hm-2 NPK-15:15:15 complex fertilizer) |
Black soil, pH:5.6 ; Total N (g/kg):1.14; Total P (g/kg): 0.31; Available N (mg/kg): 81.91; Available P (mg/kg) :38.08; Available K (mg/kg);160.00; Organic matter (g/ kg):18.86. |
K2 |
100 mg·L−1 foliar spraying |
Normal basal fertilizing |
|
K3 |
No use |
Rich basal fertilizing (750 kg·hm-2 NPK-15:15:15 complex fertilizer and 225 kg K2SO4) |
|
K4 |
100 mg·L−1 foliar spraying |
Rich basal fertilizing |
The climate : Anhui Province is a transitional region between warm temperate zone and subtropical zone in climate. In the north of Huaihe River, it belongs to the warm temperate semi humid monsoon climate, and in the south of Huaihe River, it belongs to the subtropical humid monsoon climate. Its main characteristics are: obvious monsoon, four distinct seasons, warm and changeable spring, concentrated summer rain, crisp autumn and cold winter. Anhui is also located in the middle latitude zone. With the recurrence of the monsoon, the precipitation has obvious seasonal changes. It is one of the regions with obvious monsoon climate.
Point 3:There is no technology of cultivation . There is no plant protection system, which significantly affects storage conditions (specific and quantitative composition of pests and diseases). It is necessary to indicate whether the varieties were dressed with pesticides or not. The characteristics of the studied varieties are not indicated: features of development, resistance to adverse environmental conditions, reproduction index, quality, etc.
Response 3: Thanks for your suggestion, we add it in materials and methods part.
The elongated-vine sweetpotato varieties (Z13, Z33 and J26) were used in this study. These varieties were good for preserved sweetpotato bar in China. Their vine was longer than other sweetpotato varieties. They were carried out in May and har-vested in October 2021 at CARS Sweetpotato Experiment Station, Hefei, An hui, China. The physical and chemical properties of soil and experiment design as Table 1 shown, the uniconazole spraying procedure was according to Duan et al [5]. We designed two uniconazole treatments: no use and 100 mg·L−1 UCZ spraying.100mg·L−1 UCZ reagent was prepared by mixing 5% commercially available wettable powder (Sichuan Guoguang Pesticide Co., Ltd., China) and distilled water were applied to the corre-sponding experimental fields. All treatments were uniformly applied using a foliar spray at a rate of 450 L of formulated solution per hectare. Meanwhile, we design two fertilizing treatments: normal basal fertilizing which was 450 kg/hm2 NPK-15:15:15 complex fertilizer, rich basal fertilizing which was 50 kg/hm2 NPK-15:15:15 complex fertilizer and 225 kg K2SO4. 10% Chlorpyrifos (Nanjing Huazhou Pesticide Co., Ltd., China) was used in the field before being cultivated. The trial was planted in three completely randomized blocks (about 50 m2), and Sweetpotato slip planting density was 5 plant per square meter.
Point 4:When describing experimental data, it is necessary to indicate the results of statistical data processing: whether the results are reliable or not.
Response 4: We add the p<0.05 or p>0.05 in the revison mauscript.
Point 5:The conclusions are very general. It is necessary to draw conclusions with concrete results.
Response 5: The conclusions were noticed that:Foliar UCZ applications generated desirable storage to a certain extent. The decay rate and weight loss rate decreased under 100 mg·L−1 UCZ spraying with normal basal ferti-lizing treatment in all varieties. It inhibited the decrease of the brown index to a certain degree and slowed down the decline rate of the L* value.

Reviewer 3 Report
Dear Editor,
The manuscript entitled "Effects of foliar application of uniconazole on the storage quality of tuberous roots in sweat potato" requires significant English corrections and rephrasing. I've tried to follow authors lines of thought, but it was difficult, which is very limiting for the revision process. However, I think the manuscript provides interesting research with valid results and it is generally well structured. I can have more scientific comments once the text is revised. Also, the figures presented have very low quality which makes it difficult to analyse them.
I’ve revised abstract, which was a very poor section from the manuscript and I have the following comments and suggestions, that can and should be taken into consideration for the rest of the manuscript:
Abstract
Authors do not explain why they are doing this study. They start by stating that “Foliar spraying of uniconazole (UCZ) can improve the yield of tuberous roots of sweet-16 potato in the growth stages.” But they don’t say why is that expectable. They also do not explain briefly what is uniconazole. And they don’t explain why is excepted that the foliar application of this product impacts the storage quality. In the Results part, authors refer to the decay rate, but of what?
As so, I recommend accept with major revision, after authors carefully check English, the manuscript text and figures quality provided.
Author Response
Response to Reviewer 3 Comments
Dear Editor and Reviewers:
On behalf of my co-authors, we are very grateful to you for giving us an opportunity to revise our manuscript. we appreciate you very much for your positive and constructive comments and suggestions on our manuscript entitled “Effects of foliar application of uniconazole on the storage quality of tuberous roots in sweetpotato” (ID: agronomy-2022126).We have studied reviewers’ comments carefully and tried our best to revise our manuscript according to the comments. The following are the responses and revisions I have made in response to the reviewers' questions and suggestions on an item-by-item basis. Thanks again to the hard work of the editor and reviewer!
Point 1: The figures presented have very low quality which makes it difficult to analyse them. Response 1: We have modified the figures, increased the quality of figures. The abstracut was changed as you noticed: Uniconazole (UCZ) as a plant growth regulator, has been extensively applied in sweetpota-to(Ipomoea batatas (L.) Lam) to increase tuberous root yield and quality. It is usually used in the production of sweet potato by foliar spray. The post-harvest storage stage is crucial for forming the quality of the sweetpotato's tuberous root. Few studies focus on the foliar spraying UCZ-affected storage quality of sweetpotato during pro-harvest storage. In order to examine the effects of foliar application of UCZ on the storage quality of tuberous root. This study mainly analyzed the influence of storage quality, with (K2 and K4) and without (K1 and K3) 100 mg·L−1 foliar spraying of UCZ, at storage period of normal fertilizing treatments (K1 and K2) and rich fertiliz-ing treatments (K3 and K4), on the storage quality of three representative sweetpotato varieties (Z13, Z33 and J26). Compared to the no-use UCZ treatments, the decay rate of K2 was the lowest in any storage time. The decay rate of all the varieties was 0.0% before 45 DAS. Only the decay rate of Z33 was increasing to 4.4 % at 60 DAS (p<0.05). The dry matter rate of K2 and K4 is still higher than K1 during 15~60 DAS in Z13 and J26 (p<0.05). UCZ foliar spraying was higher than without treatment during 30~60 DAS. In Z33, the springiness of UCZ spraying was higher than no spraying treatments at 45~60 DAS. These results indicate that foliar spraying of UCZ have no effects on the storage quality of tuberous root decreasing sharply, sometimes it kept the quality stable.
Point 2:Authors do not explain why they are doing this study. They start by stating that “Foliar spraying of uniconazole (UCZ) can improve the yield of tuberous roots of sweet-16 potato in the growth stages.” But they don’t say why is that expectable. They also do not explain briefly what is uniconazole. And they don’t explain why is excepted that the foliar application of this product impacts the storage quality. In the Results part, authors refer to the decay rate, but of what?
Response 2: UCZ was a good plant growth regulator, which can improve inhibited gibberellic acid (GA) and promoted abscisic acid (ABA) in flag leaves. Higher ABA in the tuberous roots is good for the extension of storage time, and lower GA is good for inhibiting the tuberous root sprout during storage. We assume the UCZ may improve the quality of tuberous root in sweetpotato.
Point 3: After the authors carefully check English, the manuscript text and figures quality provided.
Response 3: Thank you very much for discovering these errors. We apologize for tis grammatical problems and have corrected it based on our suggestions.
